# A comprehensive analysis of ribonucleotide reductase subunit M2 for carcinogenesis in pan-cancer

Yong Wang[1], Rong Chen[2], Jing Zhang[2], Peng Zeng[2]*

1 Center of Interventional Radiology and Vascular Surgery, Department of Radiology, Zhongda Hospital, Medical School, Southeast University, Nanjing, Jiangsu, China, 2 Department of Oncology, Zhongda Hospital, Medical School, Southeast University, Nanjing, Jiangsu, China

* zpeng701@163.com

## Abstract

### Background

Although there is evidence that ribonucleotide reductase subunit M2 (*RRM2*) is associated with numerous cancers, pan-cancer analysis has seldom been conducted. This study aimed to explore the potential carcinogenesis of *RRM2* in pan-cancer using datasets from The Cancer Genome Atlas (TCGA).

### Methods

Data from the UCSC Xena database were analyzed to investigate the differential expression of *RRM2* across multiple cancer types. Clinical data such as age, race, sex, tumor stage, and status were acquired to analyze the influence of *RRM2* on the clinical characteristics of the patients. The role of *RRM2* in the onset and progression of multiple cancers has been examined in terms of genetic changes at the molecular level, including tumor mutational burden (TMB), microsatellite instability (MSI), biological pathway changes, and the immune microenvironment.

### Results

*RRM2* was highly expressed in most cancers, and there was an obvious correlation between *RRM2* expression and patient prognosis. *RRM2* expression is associated with the infiltration of diverse immune and endothelial cells, immune checkpoints, tumor mutational burden (TMB), and microsatellite instability (MSI). Moreover, the cell cycle is involved in the functional mechanisms of *RRM2*.

### Conclusions

Our pan-cancer study provides a comprehensive understanding of the carcinogenesis of *RRM2* in various tumors.

**Data Availability Statement:** All relevant data are within the paper and its Supporting Information files.

**Funding:** This work was supported by the Jiangsu Provincial Special Program of Medical Science

(BE2019750), National Natural Science Foundation of China (81827805), National Key Research and Development Program (2018YFA0704100, 2018YFA0704104), Science Fund for Creative Research Groups of the National Natural Science Foundation of China (61821002), and Interventional Radiology Scientific Research Special Fund Project of Jiangsu Medical Association (SYH-3201140-0025 (2021020)).

**Competing interests:** The authors have declared that no competing interests exist.

## Introduction

Given the complex causes of tumorigenesis such as genetic loss or alteration [1] and immune infiltration [2], it is crucial to perform pan-cancer analyses of genes that may have potential clinical value. Recently, with deeper research examining the initiation, development, and treatment of tumors, many therapeutic concepts have emerged, including immune checkpoint proteins [3] and immune infiltration [4]. Interestingly, the publicly funded TGCA project and many powerful bioinformatics tools have allowed us to perform pan-cancer analyses.

Ribonucleotide reductase subunit M2 (*RRM2*), a subunit of ribonucleotide reductase, supports DNA synthesis and repair by catalyzing the formation of deoxyribonucleotides (dNTP) [5]. *RRM2* governs nucleotide metabolism in tumor cells [6] and is associated with multiple cancers [7, 8]. Several studies have demonstrated that *RRM2* is a potential diagnostic or prognostic biomarker for Ewing sarcoma [9], liver cancer [10], glioma [11], and lung adenocarcinoma [12]. Moreover, *RRM2* is an important target of many factors and drugs that suppress tumorigenicity in various cancers [13–16]. For example, DHS (trans-4,4'-dihydroxystilbene) suppresses DNA replication and tumor growth by inhibiting *RRM2* expression in pancreatic, ovarian, and colorectal cancer [17].

Although the oncogenic role of *RRM2* has been elucidated in several tumors, there is no pan-cancer evidence regarding the relationship between *RRM2* and various cancer types based on big clinical data. In this study, we performed a pan-cancer analysis of *RRM2*. We also explored the molecular mechanisms of *RRM2* in the context of different cancers using a series of factors, including gene expression, clinical correlation, survival analysis, genetic alterations, immune infiltration, immune checkpoints, enrichment analysis, and potential drug prediction.

## Materials and methods

### 1. The expression of *RRM2* in pan-cancer

To explore *RRM2* expression in different cancers, Oncomine (https://www.oncomine.org/) was used with the default parameters. Additionally, we downloaded gene expression RNA-seq data of 33 cancers from The Cancer Genome Atlas (TCGA) dataset in UCSC Xena [18]. The Perl language (https://www.perl.org/) was used to extract *RRM2* expression data from 33 cancer samples. R language was used for further analysis and visualization.

### 2. *RRM2* expression and its clinical correlation in pan-cancer

To explore the correlation between *RRM2* expression and the clinical characteristics of patients with tumors (tumor stage and survival outcome), we downloaded pan-cancer clinical data from the TGCA database (https://www.cancer.gov/). R language with limma (version 3.38.3) [19] was used for visualization.

### 3. Survival prognosis analysis and Cox regression models

Survival data for 33 cancers were downloaded from the TGCA database for survival analysis. Kaplan–Meier (KM) and Cox regression models were used to explore *RRM2* potential prognostic value of in pan-cancer. To estimate the prognostic value of *RRM2* comprehensively, we performed overall survival (years) (OS), disease-free interval (years) (DFI), disease-specific survival (years) (DSS), and progression-free interval (years) (PFI) analyses using R language with limma, survival (version 3.1.12), forestplot (version 1.9), and survminer packages (version 0.4.7).

## 4. Genetic alteration analysis

We selected the "TCGA Pan Cancer Atlas Studies" in the "quick select" section and entered "*RRM2*" (NC_000002.12NC_000002.12) for queries regarding the genetic alteration characteristics of *RRM2* in the cBioPortal web (https://www.cbioportal.org/). In the "Cancer Types Summary" module, we obtained the alteration frequency, mutation data (the type of mutation tumors), and CNA data (copy number alteration) of *RRM2* in all TGCA tumors. In the "Mutations" module, the mutated site information of *RRM2* was visualized in a schematic diagram of the protein structure or the 3D structure. We used the R language with the fmsb package to explore the correlation between *RRM2* expression and tumor mutations in 33 TGCA cancers downloaded from the TGCA database.

Tumor mutational burden (TMB) is a biomarker that represents the total number of mutations in a tumor [20]. Microsatellite instability (MSI) reflects the loss or gain of nucleotides from repetitive DNA tracts and is a diagnostic phenotype of many cancers [21]. We explored the correlation between *RRM2* expression and TMB and also MSI in pan-cancer using Perl and R language with the fmsb package (version 0.7.0).

## 5. Immune infiltration analysis

We chose the "Immune" module of the TIMER2 webserver [22], and RRM2 and endothelial cells were then selected for exploring the association between immune infiltrates of endothelial cells and RRM2 expression in all TCGA. A partial Spearman's correlation analysis was performed using the Purity Adjustment" option. We further investigated the correlation between RRM2 expression and the infiltration levels of the other 22 immune cells. The R language with ggplot2 (version 3.3.2), ggpubr, and ggExtra (version 0.9) packages was used for analysis and visualization.

## 6. Immune checkpoint analysis

Immune checkpoint molecules are costimulatory receptors on immune cells that regulate immune cells through an immune response [23]. In this study, we used R language with the limma, reshape2, and RColorBrewer packages to analyze the relationship between RRM2 expression and the expression levels of 47 common immune checkpoint genes in pan-cancer.

## 7. Enrichment analysis of *RRM2*-related gene

In the STRING website (https://string-db.org/), we input the protein name ("RRM2") and organism ("Homo sapiens") in the "Protein by name" module. Subsequently, we changed the basic settings as follows: meaning of network edges: evidence; active interaction sources: experiments; minimum required interaction score: low confidence (0.150); maximum number of interactors to demonstrate no more than 50 interactors in 1st shell. Eventually, RRM2-binding proteins that were authenticated by experiments were acquired. Cytoscape (version 3.6.1) was used for integration and visualization [24]. In the "Similar Genes" section of GEPIA2 (http://gepia2.cancer-pku.cn/), we searched *RRM2*-correlated targeting genes by Gene ("*RRM2*"), Top # ("100") similar Genes, and all TGCA tumor and normal tissues. Meanwhile, we used the "correlation analysis" module to explore the Pearson correlation between *RRM2* expression and the top 5 *RRM2*-correlated targeting genes in all TGCA tumor and normal tissues. The "Gene_Corr" module in TIMER2 was used to indicate the correlation between *RRM2* with the top 5 *RRM2*-correlated targeting genes in various cancer types. Subsequently, we performed intersection analysis to compare the genes with which *RRM2* binds and interacts using Jvenn [25]. Furthermore, we performed Kyoto Encyclopedia of Genes and Genomes

(KEGG) enrichment analysis for the two sets of data that were *RRM2*-binding and correlated through the R language with clusterProfiler, ggplot2, and enrichplot packages.

## 8. Potential drug prediction for pan-cancer

The KEGG gene sets as Gene Symbols were downloaded from the GESE online dataset (https://www.gsea-msigdb.org/). R language with colorspace, stringi, ggplot2, limma, and enrichplot packages was used to explore *RRM2*-related KEGG enrichment analyses in pan-cancer. Finally, enriched pathways were counted for further drug prediction. In the Therapeutic Target Database (http://db.idrblab.net/), we selected the "Pathway Search" section of the "Advanced Search" module [26]. Then, step1: KEGG and step2: CELL CYCLE were chosen to identify pathway-associated drugs related to *RRM2* expression in pan-cancer. Only approved drugs were screened.

## Results

### 1. The expression of *RRM2* in pan-cancer

To explore the oncogenic role of human *RRM2*, we analyzed the expression pattern of *RRM2* in different cancer types using the Oncomine database (Fig 1A). The results revealed that *RRM2* expression was significantly increased in most cancer groups, including bladder, brain, breast, cervical, colorectal, esophageal, gastric, head and neck, kidney, liver, lung, lymphoma, melanoma, ovarian, pancreatic, prostate, and sarcoma cancers. However, the expression of *RRM2* is decreased in leukemic cells. Furthermore, we identified the expression levels of *RRM2* in 33 different tumor and non-tumor tissues based on TCGA datasets. As presented in Fig 1B, the expression levels of *RRM2* were extremely different between the cancer and normal groups. The expression levels of *RRM2* were significantly higher in 19 tumor tissues than they

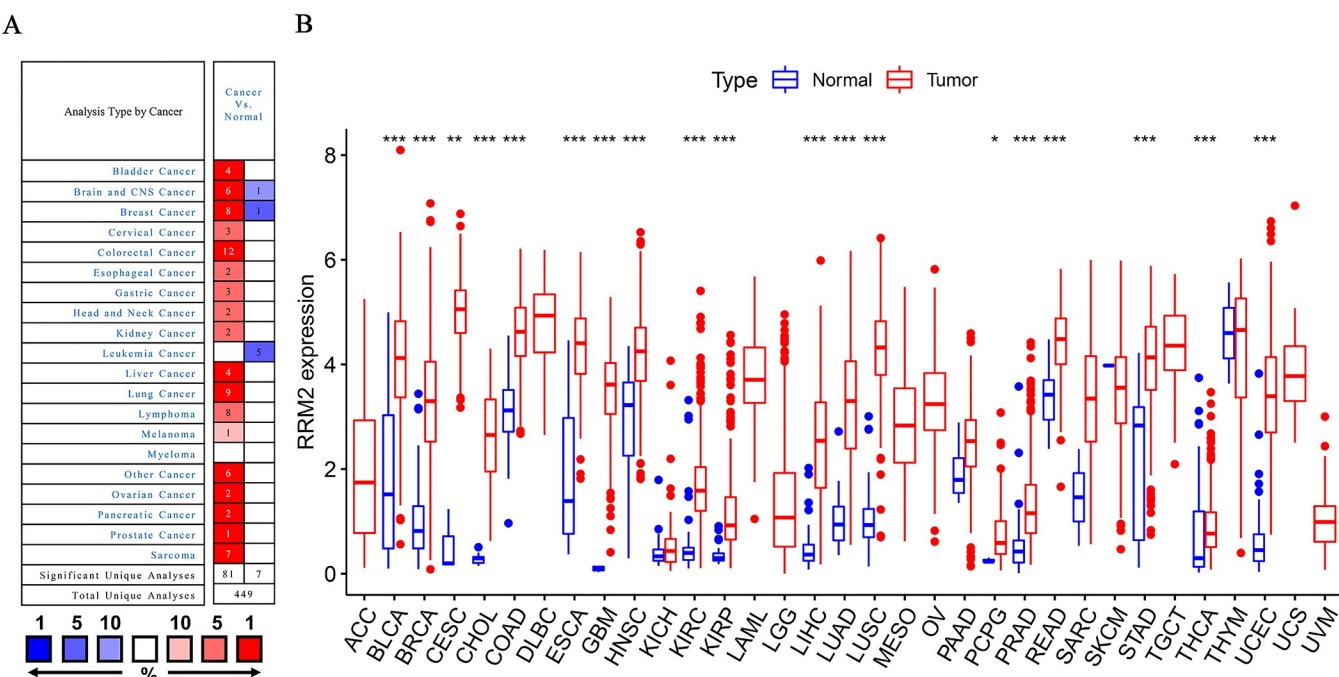

**Fig 1. The expression level of *RRM2* in pan-cancer.** (A) The expression level of *RRM2* in different pan-cancers as indicated in the Oncomine database. Gene rank percentile (%). (B) *RRM2* differential expression in pan-cancer in the TCGA database. *P < 0.05; **P < 0.01; ***P < 0.001.

were in the corresponding normal tissues, including BLCA (Bladder Urothelial Carcinoma), BRCA (breast invasive carcinoma), CESC (cervical squamous cell carcinoma and endocervical adenocarcinoma), CHOL (cholangiocarcinoma), COAD (colon adenocarcinoma), ESCA (esophageal carcinoma), GBM (glioblastoma multiforme), HNSC (Head and Neck squamous cell carcinoma), KIRC (kidney renal clear cell carcinoma), KIRP (kidney renal papillary cell carcinoma), LIHC (liver hepatocellular carcinoma), LUAD (lung adenocarcinoma), LUSC (lung squamous cell carcinoma), PCPG (Pheochromocytoma and Paraganglioma), PRAD (prostate adenocarcinoma), READ (rectal adenocarcinoma), STAD (stomach adenocarci-noma), THCA (thyroid carcinoma), and UCEC (Uterine Corpus Endometrial Carcinoma). Overall, these results indicated that *RRM2* is highly expressed in most cancers and plays a potential oncogenic role.

## 2. *RRM2* expression and clinical correlation in pan-cancer

We assessed the correlation between the expression of *RRM2* and the clinical characteristics of patients with pan-cancer, including age, race, tumor stage, and status. For age, significant cor-relations with *RRM2* expression were observed in 0–29 years group (BRCA, KIRP, LUAD, PCPG, READ, STAD, TGCT (Testicular Germ Cell Tumors)), 30–49 years group (ESCA, KIRC, LAML (Acute Myeloid Leukemia), LIHC, THYM (thymoma)), 50–70 years group (LGG (brain lower-grade glioma)), LUSC), and over 70 years group (KICH (Kidney Chromo-phobe), PAAD (pancreatic adenocarcinoma)) (Fig 2A–2D and S1A-S1L Fig in S1 File). As for race, *RRM2* expression was significantly correlated with BRCA, KICH and KIRP in asian group, BLCA, LIHC and THYM in black group, and KIRC in white group (Fig 2E–2H and S1M-S1O Fig in S1 File). Regarding the tumor stage, *RRM2* expression was significantly asso-ciated with Skin Cutaneous Melanoma (SKCM) in stage I group, BRCA, COAD, LUAD, LUSC and PAAD in stage II group, KIRP, LIHC and THCA in stage III group, adrenocortical carcinoma (ACC), KICH, KIRC and TGCT in stage IV group (Fig 2I–2L and S1P-S1X Fig in S1 File). In terms of tumor status, *RRM2* expression levels were significantly correlated with COAD and OV (ovarian serous cystadenocarcinoma) in tumor free group, ACC, BLCA, KICH, KIRC, KIRP, LGG, LUAD, PAAD, PCPG, PRAD and UVM (Uveal Melanoma) in tumor group (Fig 2M–2P and S1Y-S1AG Fig in S1 File).

## 3. Survival analysis

To estimate the correlation between *RRM2* expression and pan-cancer prognosis, we divided the tumor cases into two groups based on high and low expression levels of *RRM2*. The Kaplan–Meier method was applied to analyze the overall survival (years) (OS), disease-free interval (years) (DFI), disease-specific survival (years) (DSS), and progression-free interval (years) (PFI). As presented in Fig 3A–3E and S2A-S2I Fig in S1 File, OS analysis data revealed high *RRM2* expression was associated with poor prognosis for the TCGA cases of ACC (P<0.001), KICH (P = 0.015), KIRC (P<0.001), KIRP (P<0.001), LGG (P<0.001), LIHC (P = 0.005), LUAD (P<0.001), MESO (Mesothelioma) (P<0.001), PAAD (P = 0.006), PRAD (P = 0.004), SARC (Sarcoma) (P = 0.04), UCEC (P = 0.021), and UVM (P<0.001). However, low *RRM2* expression was associated with poor OS in patients with THYM (P = 0.017). DSS analysis data indicated a correlation between high *RRM2* expression and poor prognosis in TGCA cases of ACC (P<0.001), KICH (P = 0.001), KIRC (P<0.001), KIRP (P<0.001), LGG (P<0.001), LIHC (P = 0.028), LUAD (P = 0.004), MESO (P<0.001), PAAD (P = 0.014), PRAD (P = 0.03), and UVM (P<0.001) (Fig 3F–3J and S2J-S2O Fig in S1 File). For DFI (Fig 3K–3O and S2P-S2R Fig in S1 File), high *RRM2* expression was linked to poor prognosis for cancers of KIRP (P = 0.007), LIHC (P = 0.038), LUAD (P = 0.005), PAAD (P = 0.022), SARC

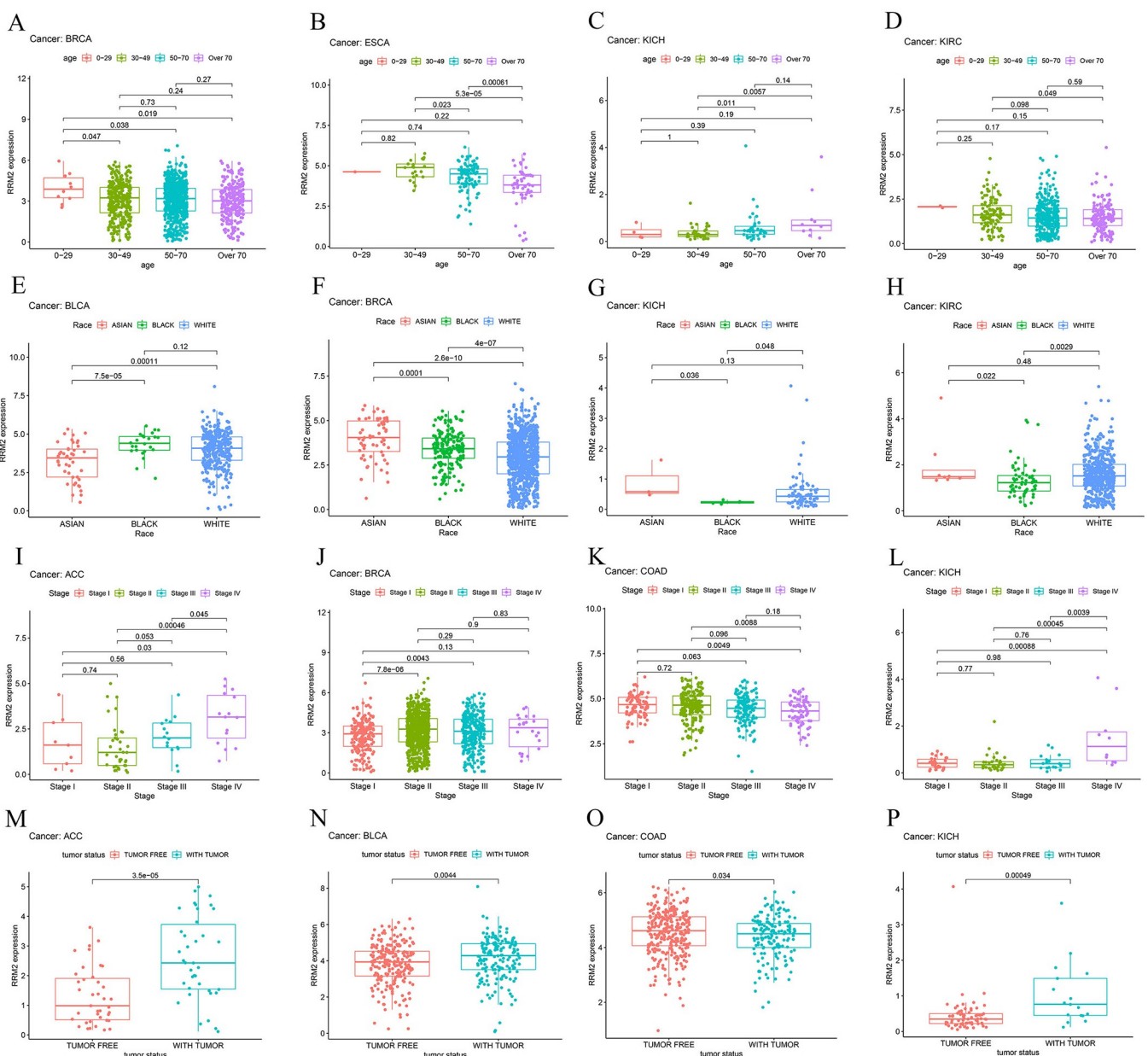

**Fig 2. The expression level of *RRM2* and its clinical correlation in pan-cancer.** (A-D) The clinical correlation between *RRM2* expression level and age in BRCA, ESCA KICH, and KIRC, respectively. (E-H) The clinical correlation between *RRM2* expression level and race in BLCA, BRCA, KICH, and KIRC, respectively. (I-L) The clinical correlation between *RRM2* expression level and tumor stage of patients in ACC, BRCA, COAD, and KICH, respectively. (M-P) The clinical correlation between *RRM2* expression level and tumor status of patients in ACC, BLCA, COAD, and KICH, respectively. The number above the horizontal line represents the p-value between the two groups.

(P<0.001), TGCT (P = 0.043) and THCA (P = 0.001). Additionally, low *RRM2* expression was associated with poor DFI for OV (P = 0.022). For PFI analysis data (Fig 3P–3T and S2A, S2B Fig in S1 File), high *RRM2* expression was correlated with poor prognosis for ACC (P = 0.002), KICH (P = 0.007), KIRC (P = 0.025), KIRP (P<0.001), LGG (P = 0.001), LIHC (P = 0.003), LUAD (P = 0.003), MESO (P<0.001), PAAD (P = 0.002), PRAD (P<0.001), SARC (P = 0.001), THCA (P = 0.011), and UVM (P<0.001). However, low *RRM2* expression was associated with poor PFI in COAD (P = 0.049) and STAD (P = 0.05).

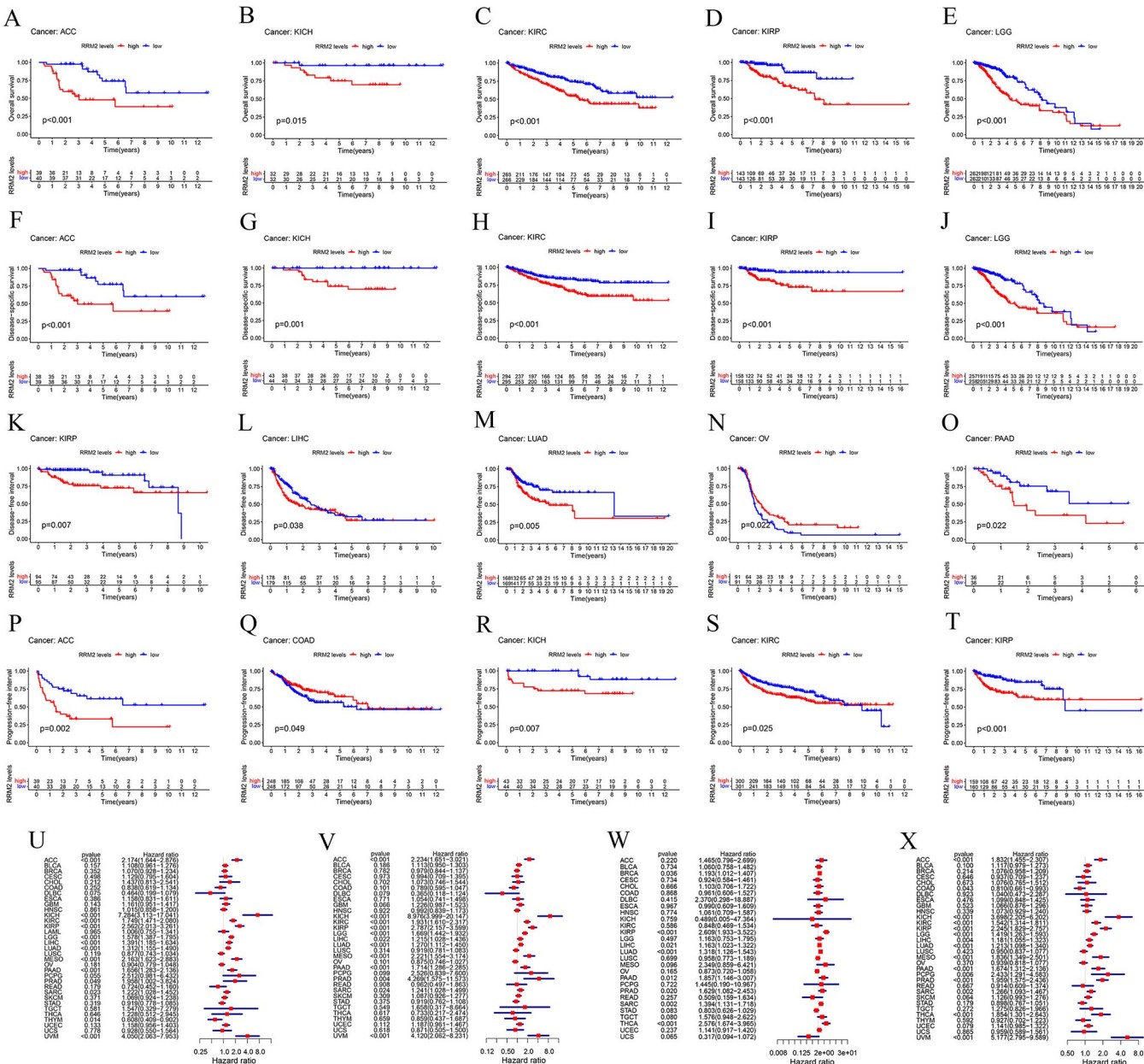

**Fig 3. Prognostic value of *RRM2* in pan-cancer.** (A-E) Correlation between high expression level of *RRM2* and poor OS in ACC, KICH, KIRC, KIRP, and LGG using the Kaplan–Meier method. (F-J) Correlation between high expression level of *RRM2* and poor DSS in ACC, KICH, KIRC, KIRP, and LGG using the Kaplan–Meier method. (K-O) Correlation between high expression level of *RRM2* and poor DFI in KIRP, LIHC, LUAD, PAAD, and SARC using the Kaplan–Meier method. (P-T) Correlation between high expression level of *RRM2* and poor PFI in ACC, KICH, KIRC, KIRP, and LGG using the Kaplan–Meier method. (U-X) Correlation between *RRM2* expression and OS, DSS, DFI, and PFI in pan-cancer using the Cox regression model. OS, Overall survival (years); DSS, Disease-specific survival (years); DFI, Disease-free interval (years); PFI, Progression-free interval (years).

Furthermore, we calculated survival data using Cox regression models (Fig 3U–3X). The OS, DSS, and PFI results were similar to those calculated using the Kaplan–Meier method. The difference in OS was that high *RRM2* expression was not significantly correlated with poor prognosis in UCEC. Additionally, high *RRM2* expression was associated with poor DSS in patients with SARC (P = 0.024). For PFI, high *RRM2* expression was significantly correlated with poor prognosis for PCPG (P = 0.006) but was not significantly correlated with poor

prognosis for STAD. For DFI, the data indicated that the high *RRM2* expression was associated with the poor prognosis for cancers that included BRCA (P = 0.036), KIRP (P<0.001), LIHC (P = 0.021), LUAD (P<0.001), PAAD (P = 0.012), PRAD (P = 0.020), SARC (P = 0.002), and THCA (P<0.001). These data suggest that *RRM2* is differentially associated with the prognosis of various cancers.

## 4. Genetic alteration analysis data

Next, we explored the genetic alteration status of *RRM2* in 10,953 patients/10,967 samples from 32 TGCA cohort studies. The analysis data revealed that *RRM2* exhibited the highest alteration frequency (>7%) in patients with uterine carcinosarcoma, among which "amplification" was the main type. Notably, the genomic alteration types in all diffuse large B-cell lymphoma (DLBCL) and KICH cases were deep deletions in *RRM2* (Fig 4A). We investigated the types, sites, and numbers of *RRM2* mutations (Fig 4B). The primary genetic mutation in *RRM2* was a missense mutation. The R298Q/W mutation in ribonucleotide reductase that was detected in three cases of UCEC and one case of GBM was able to produce a missense mutation in the *RRM2* gene, thus causing translation from R (arginine) to W (tryptophan) or glutamine (Q) at position 298 of the RRM2 protein. We visualized the R298 site in the 3D structure of RRM2 (Fig 4C).

Moreover, we explored the correlation between *RRM2* expression and tumor mutational burden (TMB) and microsatellite instability (MSI) in all TCGA cancers. As presented in Fig 4D, *RRM2* expression was positively correlated with TMB in ACC, BLCA, BRCA, CESC, CHOL, COAD, KICH, KIRC, LGG, LIHC, LUAD, LUSC, MESO, OV, PAAD, PRAD, READ, SARC, SKCM, and STAD but negatively correlated with THYM. As presented in Fig 4E, we observed a positive correlation between *RRM2* expression and MSI for COAD, LIHC, SARC, STAD, TGCT, UCEC, and UCS (Uterine Carcinosarcoma) but a negative correlation for LAML and SKCM. These results should be subjected to more in-depth analysis.

## 5. Immune infiltration

Immune cells, the primary components of the tumor microenvironment (TME), are involved in the progression and metastasis of tumors and therapy resistance [23, 27]. Tumor-associated endothelial cells have been reported to play a key role in sculpting the immune responses necessary for tumor growth and metastasis [28]. First, we explored the association between *RRM2* expression and the infiltration level of immune cells in TCGA cancers using the TIMER, CIBERSORT, CIBERSORT-ABS, QUANTISEQ, XCELL, MCPCOUNTER, and EPIC algorithms. As presented in Fig 5A, *RRM2* expression levels in BRCA, KIRC, LUAD, LUSC, STAD, THCA, and THYM were negatively correlated with the infiltration level of tumor-associated endothelial cells but were positively correlated with KIRP and LGG. We then analyzed the correlation between *RRM2* expression levels and the immune infiltration of different tumor-associated cells (B cells, plasma cells, T cells, NK cells, monocytes, macrophages, dendritic cells, mast cells, eosinophils, and neutrophils) and their subtypes in tumors. The results indicated that *RRM2* expression levels were significantly correlated with the immune infiltration of various tumor-associated cells in 24 tumor types. For example, *RRM2* expression in the ACC was negatively correlated with the infiltration of resting mast cells (Fig 5B). In BLCA, *RRM2* expression significantly correlated with the infiltration of five types of immune cells (Fig 5C–5G). In BRCA, *RRM2* expression was significantly correlated with the infiltration of 10 types of immune cells (Fig 5H–5Q). The correlation between *RRM2* expression and immune cell infiltration in other cancers is presented in S3A-S3CI Fig in S1 File.

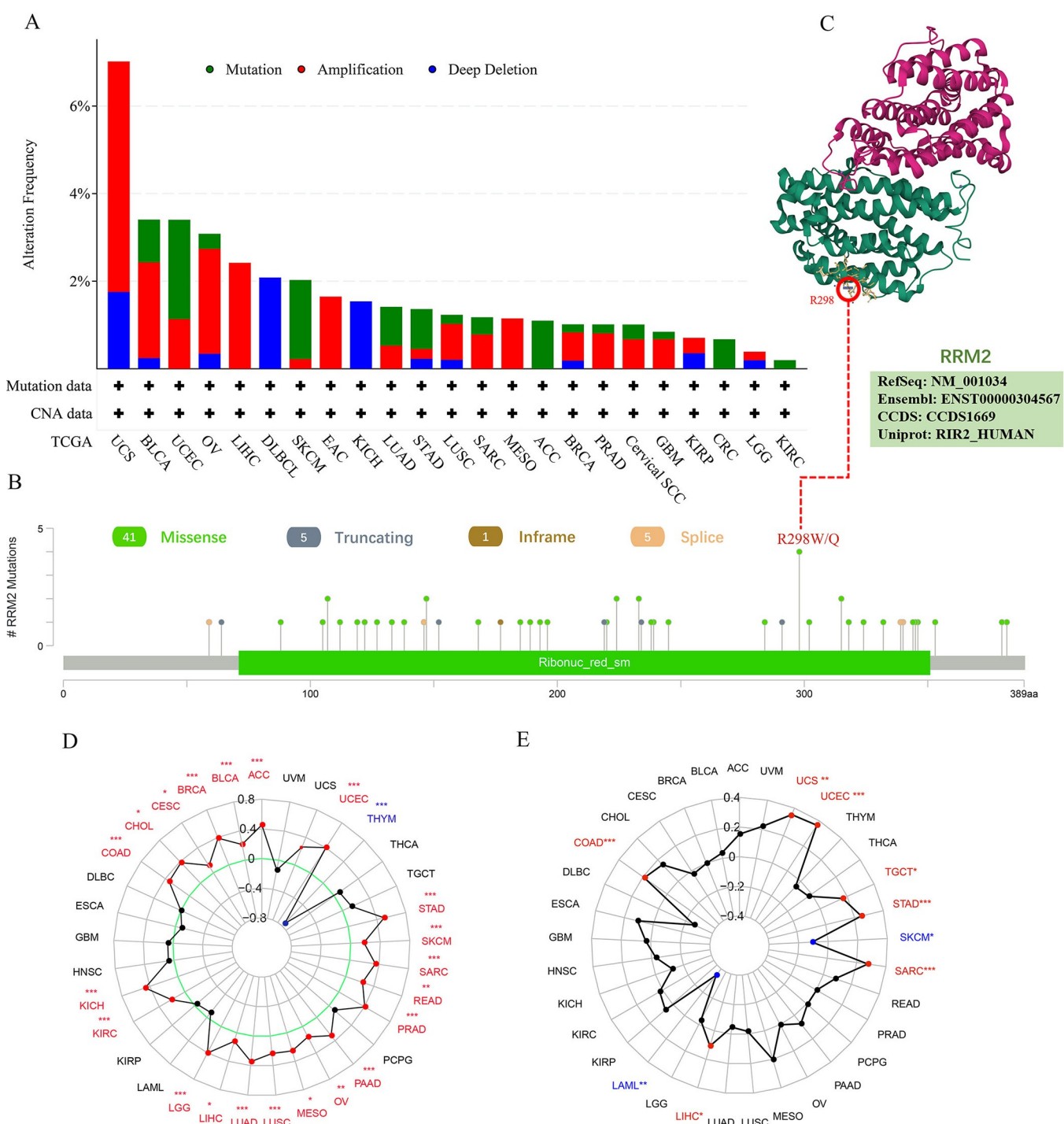

**Fig 4. Mutation features of *RRM2* in different tumors of TCGA.** (A, B) The alteration frequency of *RRM2* with mutation type and mutation site. (C) The mutation site with the highest alteration frequency (R298Q/W) in the 3D structure of *RRM2*. (D) A radar map was used to reflect the correlation between *RRM2* expression and TMB. (E) A radar map was used to reflect the correlation between *RRM2* expression and MSI. TMB, Tumor mutational burden; MSI, Microsatellite instability.

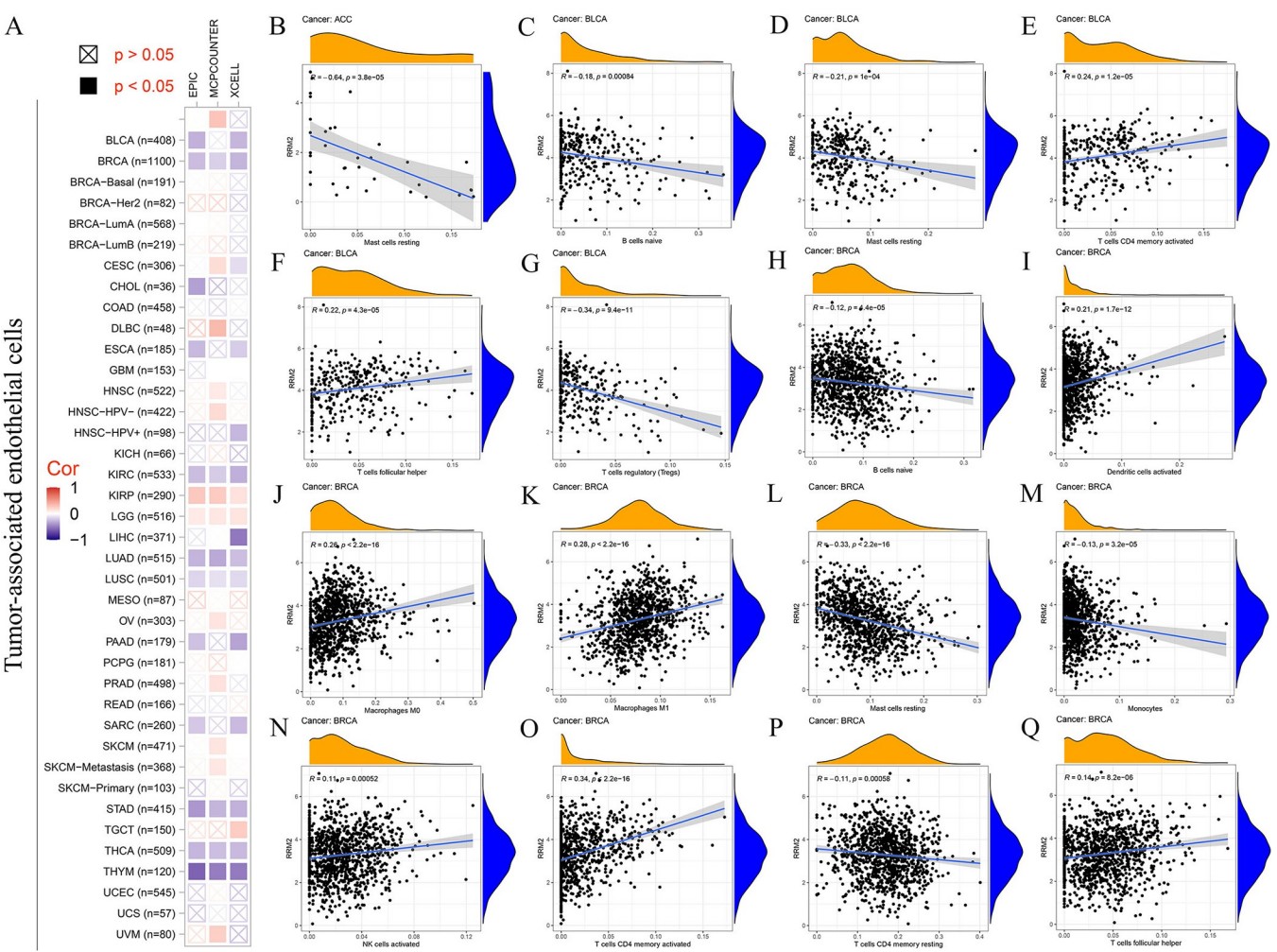

**Fig 5. Correlation analysis between *RRM2* expression and immune infiltration of cancer-associated cells.** (A) The correlation between *RRM2* gene expression and the infiltration level of endothelial cells in all types of cancer in TCGA. (B-Q) The correlation between *RRM2* gene expression and the infiltration level of diverse immune cells in ACC, BLCA, and BRCA, respectively.

## 6. Immune checkpoint analysis

We explored the correlation between *RRM2* and immune checkpoint gene expression in various cancers (Fig 6). The results revealed that the expression levels of more than 40 immune checkpoint genes were significantly associated with *RRM2* expression levels in TGCT and THCA. Moreover, the expression levels of up to 30 immune checkpoint genes in BRCA, HNSC, KIRC, KIRP, LGG, LIHC, PRAD, THYM, and UVM correlated with *RRM2* expression levels. Additionally, more than 20 immune checkpoint genes were associated with *RRM2* expression in 10 cancer types.

## 7. Enrichment analysis of *RRM2*-related gene

To understand the molecular mechanism of action of RRM2 in the context of tumorigenesis, we searched for *RRM2* expression-correlated target genes and RRM2-binding proteins for functional enrichment analysis. We identified 50 RRM2-binding proteins that were authenticated using STRING. The protein-protein interaction network is presented in Fig 7A. We then

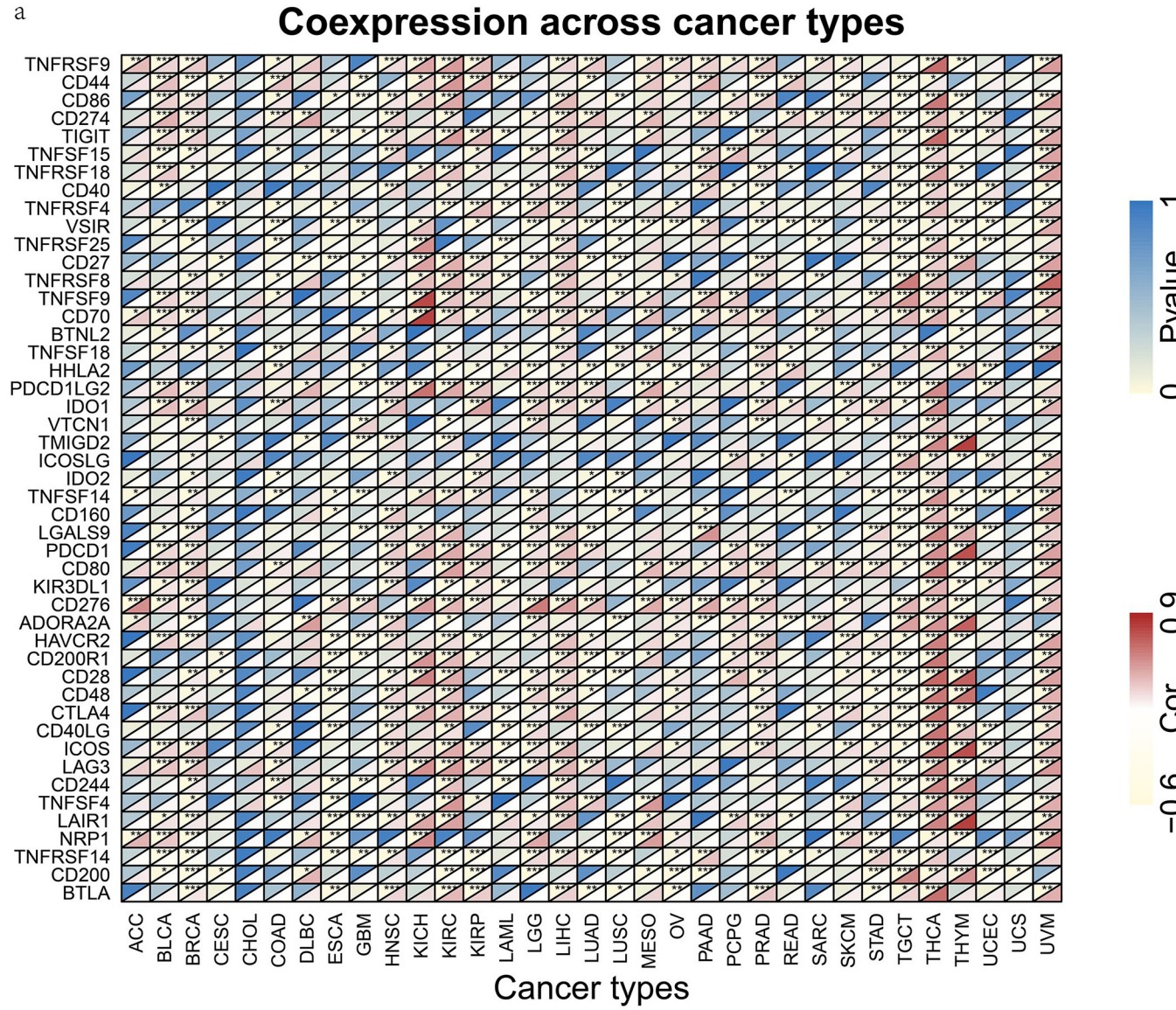

**Fig 6. RRM2 expression level and immune checkpoint genes in pan-cancer.** (A) Heatmap of the correlation between *RRM2* expression and immune checkpoint genes.

explored the top 100 *RRM2* expression-correlated genes based on the TGCA datasets using GEPIA2. The results indicated that the expression level of *RRM2* was positively associated with that of MKI67 (marker of proliferation Ki-67) (R = 0.79), ORC1 (origin recognition complex subunit 1) (R = 0.78), CCNA2 (Cyclin A2) (R = 0.77), PLK1 (polo-like kinase 1) (R = 0.77), and KIF11 (kinesin family member 11) (R = 0.76) (Fig 7B). The heatmap indicated a positive correlation between *RRM2* and the four genes in virtually all cancer types (Fig 7C). The above two groups possessed four common genes that included PLK1, CDK1 (cyclin-dependent kinase 1), DTL (Denticleless E3 ubiquitin protein ligase homolog), and ASF1B (anti-silencing function 1 B histone chaperone) (Fig 7D). KEGG enrichment analyses were performed based on the genes in the two groups. The result indicated that "cell cycle" may play a crucial role in the effect of *RRM2* on tumor pathogenesis (Fig 7E).

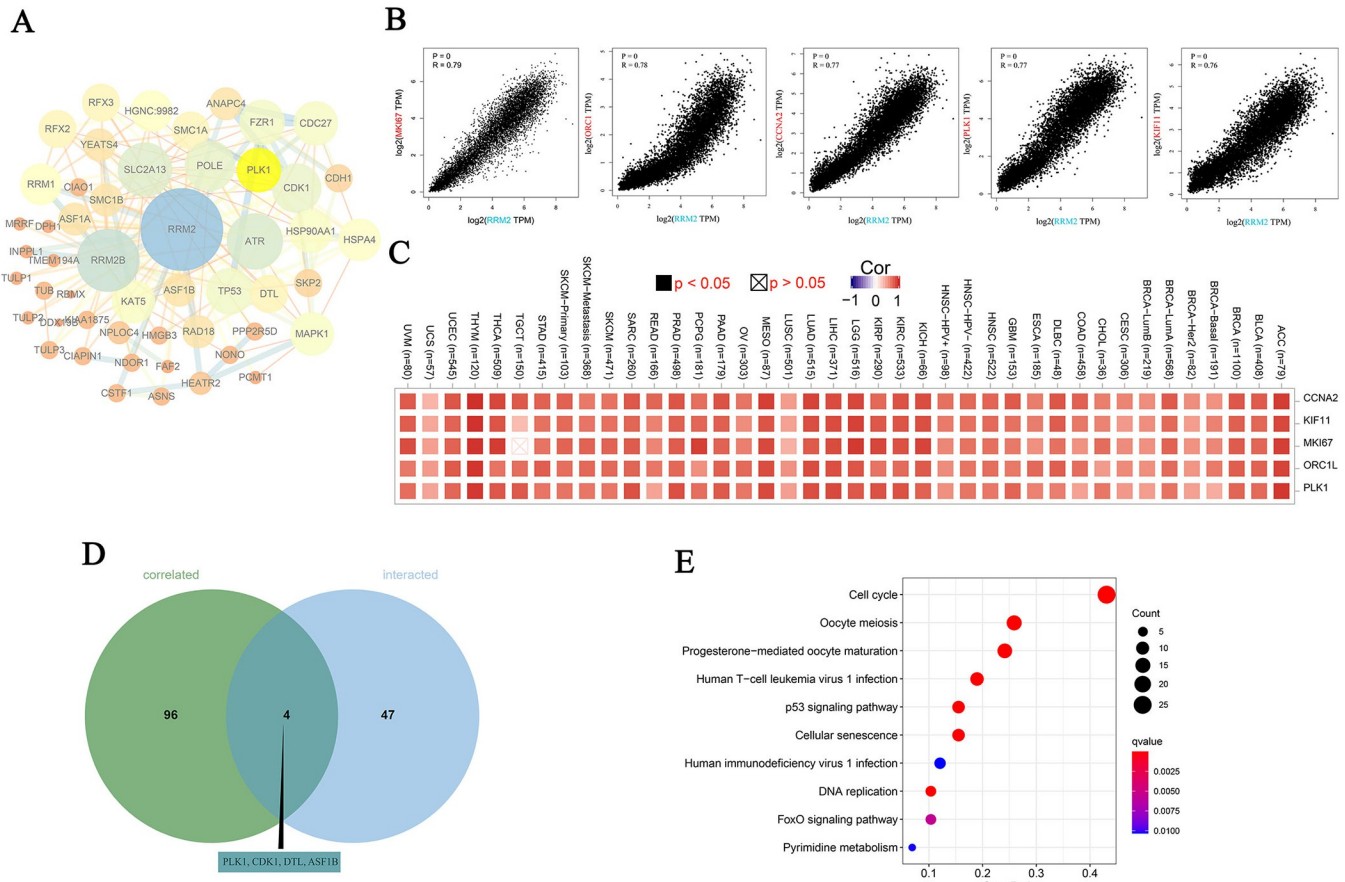

**Fig 7. *RRM2*-related gene enrichment analysis.** (A) Protein-protein interaction network of the available experimentally determined RRM2 binding proteins. The size and color of the node depends on the degree. The width of the edge is determined by the combined score of STRING. (B) The expression correlation between *RRM2* and the top 5 *RRM2*-correlated genes in TCGA, including MKI67, ORC1, CCNA2, PLK1 and KIF11. (C) The corresponding heatmap data for the detailed cancer types. (D) An intersection analysis of the *RRM2*-binding and correlated genes. (E) KEGG pathway enrichment analyses of the genes that *RRM2*-binding and interacted.

## 8. Potential drug prediction for pan-cancer

We performed a gene set enrichment analysis (GSEA) to explore the pathways regulated by *RRM2* in pan-cancer cells (S4 Fig in S1 File). The results revealed that the cell cycle pathway was the most significantly affected by *RRM2* expression in pan-cancer, and this was consistent with the KEGG enrichment analyses (S1 Table in S1 File). Drug prediction analysis indicated that up to 14 approved drugs such as Phenylbutyrate and Romidepsin may exert pan-cancer therapeutic effects by targeting cell cycle-associated pathways (S2 Table in S1 File).

## Discussion

Consistent with the extremely important biological functions of *RRM2* in DNA replication [29], *RRM2* has been demonstrated to exhibit a functional link with diverse tumors [7–9]. Whether *RRM2* is involved in the pathogenesis of multiple tumors via a common molecular mechanism remains unclear. Through bibliography retrieval, we failed to identify any publication with a pan-cancer analysis of *RRM2* based on the perspective of all tumors. Therefore, in this study, we uncovered for the first time the role of *RRM2* in 33 different cancers. Using bioinformatic analysis of the Oncomine and TCGA public databases, we demonstrated that the

expression of *RRM2* in up to 19 tumor tissues was higher than that in the corresponding normal tissues. These findings demonstrated that *RRM2* plays an important role in most tumors (Fig 1). Subsequently, the significance of *RRM2* in pan-cancer clinical prognosis was explored. The expression of *RRM2* was significantly related to age, race, tumor stage, and status in nearly one-third of the TGCA (Fig 2). Additionally, survival analyses using Kaplan–Meier and Cox regression models revealed that most patients with cancer and high *RRM2* expression exhibited significantly worse survival rates than those with low expression (Fig 3). Considering the patient samples in the cBioPortal database, *RRM2* gene mutations appeared in 23 types of patients with tumors, with an alteration frequency of 0.2% to 7.02%. We observed that most alterations in *RRM2* were gene mutations, amplifications, and deep deletions (Fig 4). We also observed evidence of a close association between *RRM2* expression and TMB or MSI in TCGA samples. These results demonstrate that *RRM2* may be a powerful prognostic biomarker in pan-cancer and may expedite the development of precise targeted therapies for tumors.

*RRM2* is highly expressed in the majority of tumors. However, given the limitations of online databases and tools, the expression of *RRM2* in certain tumors was omitted. We supplemented missing meaningful results with a literature search. Tabbal et al. demonstrated that high expression of *RRM2* induced by EZH2 is associated with poor prognosis in adrenocortical carcinoma (ACC) [30]. Inhibition of *RRM2* can block cell proliferation and migration and induce apoptosis, thus suggesting that *RRM2* may be a target in ACC [30]. Aimiuwu et al. demonstrated that 5-Azacytidine (5-azaC) could cause perturbation of deoxyribonucleotide pools (LAML) and that *RRM2* may be a novel molecular target of 5-azaC in AML [31]. In Mesothelioma, Wendy et al. reported that *RRM2* was highly expressed in MESO by semi-quantitative immunohistochemical analysis in 70 patients, but no significant correlation was observed between *RRM2* expression and the clinical prognosis of OS and PFS [32]. The current data revealed a strong relationship between *RRM2* and up to 20 types of tumors; however, further research is required to explore if *RRM2* is related to the remaining tumors. Consistent with this result, our clinical data correlation analysis indicated that *RRM2* was strongly correlated with age, race, tumor stage, and cancer status. These results indicate that *RRM2* plays an important pan-cancer role and possesses clinical value that is worthy of in-depth investigation.

Cox regression models and the Kaplan–Meier method were used to evaluate the survival of patients with *RRM2* tumors. However, for certain tumors, the results of the two algorithms were inconsistent. In this study, we used Cox regression survival analysis of data from the TCGA-UCEC cohort and observed a statistical correlation between high *RRM2* expression and poor OS (P = 0.021). Nevertheless, no significant correlation was observed between *RRM2* expression and OS in UCEC using the Kaplan–Meier method (P = 0.133). Different data-processing methods or updated survival data may have contributed to these results. Therefore, we used another web tool, GEPIA2 [33], and failed to detect a correlation between *RRM2* expression and overall survival (years) of patients with UCEC (P = 0.064). Consequently, the current evidence cannot confirm the role of *RRM2* expression in the OS of UCEC. For other tumors with identical DFI, DSS, and PFI results, larger sample sizes and clinical studies are needed to further confirm the relationship between *RRM2* and the survival prognosis of different tumors, and this indicates the complexity of tumors and the diverse roles of *RRM2*.

For pancreatic adenocarcinoma (PAAD), we analyzed the TCGA-PAAD dataset (n = 183) and failed to observe significant differential expression of *RRM2* in PAAD tissues compared to levels in normal tissues. However, a significant correlation was observed between *RRM2* expression and the clinical prognoses of OS, PFS, RFS, and DSS in patients with PAAD. A recent study by Anna et al. was in agreement with the results of the survival analysis [34]. Therefore, larger sample sizes are required to confirm the role of *RRM2* in PAAD.

In this study, we demonstrated that pan-cancer *RRM2* expression levels correlated with the immune infiltration levels of diverse tumor-associated cells (Fig 5). High *RRM2* expression leads to decreased immune infiltration in most cancers, including ACC, KIRC, KIRP, LGG, LIHC, LUAD, PAAD, PRAD, SARC, TGCT, THCA, UCEC, and UVM. Consistent with this finding, high RRM2 expression was associated with poor prognosis in these tumors. These findings indicate that in at least 13 types of tumors, *RRM2* can affect tumor prognosis by regulating the level of immune infiltration.

Immune checkpoint blockade has exhibited remarkable advantages in the treatment of several cancer types [35]. Immune checkpoint blockade increases anti-tumor immunity [35]. Here, we demonstrate that *RRM2* expression is closely related to immune checkpoint genes in various cancers (Fig 6). In renal cell carcinoma, Xiong et al. reported that *RRM2* promotes sunitinib resistance and that knockdown of *RRM2* enhances the anti-tumor efficiency of programmed cell death protein 1 (PD-1) blockade in renal cancer [36]. Our data analysis also revealed that the expression of *RRM2* was related to 21, 38, and 30 checkpoint genes in KICH, KIRC, and KIRP cells, respectively. Additionally, we confirmed that the expression of *RRM2* is positively associated with PDCD1 (PD-1) in KICH, KIRC, and KIRP. These results suggested that *RRM2* is a promising therapeutic target for renal cell carcinoma. Nevertheless, the specific immunotherapeutic role of *RRM2* in immune checkpoint blockade requires further pan-cancer studies.

First, we integrated the data for *RRM2*-binding components and *RRM2* expression-correlated target genes across all tumors (Fig 7). KEGG and KSEA enrichment analyses indicated that "cell cycle," "oocyte meiosis," "p53 signaling pathway," "cell senescence," and "DNA replication" play potential roles in the effect of *RRM2* on tumor pathogenesis. One research indicates that cell cycle dependent *RRM2* may serve as proliferation marker and pharmaceutical target in adrenocortical cancer [37]. Chao Ma et. al displayed *RRM2* was an independent prognostic factor in lung adenocarcinoma which was closely related oocyte meiosis [38]. Mingxue Yu et. al found that *RRM2* were enriched in the p53 signaling pathway and could be potential biomarkers and therapeutic targets for HBV-related HCC [39]. Letizia Granieri et. al revealed that targeting the USP7/RRM2 axis drives senescence and sensitizes melanoma cells to HDAC/LSD1 inhibitors [40]. Wenxiu Qi et. al suggested that VE-822 and AZD1775 decreased the protein levels of ribonucleotide reductase M1 (RRM1) and M2 (RRM2) subunits, key enzymes in the synthesis of deoxyribonucleoside triphosphate, which increased DNA replication stress [41].

Given the importance of *RRM2* in tumors, we explored drugs that target *RRM2*. We identified drugs that target the cell cycle pathway most affected by *RRM2* expression in pan-cancers. Nevertheless, whether these drugs play a role in the treatment of cancer or certain tumors must be further explored. *RRM2* converts the active metabolite of gemcitabine into its inactive form [42]. Moreover, alterations in *RRM2* are associated with resistance to gemcitabine-induced cell death in lung cancer and pancreatic ductal adenocarcinoma [42]. Therefore, we speculated that some of these predicted drugs could regulate the efficacy of gemcitabine in certain cancers. Further research is required to confirm these findings.

In summary, our first pan-cancer analysis of *RRM2* revealed statistical correlations between *RRM2* expression and clinical prognosis, genetic alterations, tumor mutational burden, microsatellite instability, immune cell infiltration, and immune checkpoints across multiple tumors. This helped us to understand the role of *RRM2* in tumorigenesis from the perspective of clinical tumor samples.

## Supporting information

**S1 File. Contains all data for S1–S4 Figs and S1, S2 Tables.**
(DOCX)

## Acknowledgments

Special thanks to Yong Wang for performing the data analysis. We are also grateful for the data collection by Jing Zhang. All authors contributed equally to data analysis, experiments, and overall manuscript conception.

This work has been uploaded in a preprint. We thank Research Square for presenting and evaluating our study (https://www.researchsquare.com/article/rs-2215057/v1).

## Author Contributions

**Conceptualization:** Peng Zeng.

**Data curation:** Yong Wang.

**Formal analysis:** Jing Zhang.

**Project administration:** Peng Zeng.

**Writing – original draft:** Rong Chen.

**Writing – review & editing:** Peng Zeng.

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
