## [Decision Letter · Decision Letter 0]

18 Jan 2024

PONE-D-23-34369A Comprehensive Analysis of Ribonucleotide Reductase Subunit M2 to Carcinogenesis in Pan-CancerPLOS ONE

Dear Dr. Zeng,

Thank you for submitting your manuscript to PLOS ONE. After careful consideration, we feel that it has merit but does not fully meet PLOS ONE’s publication criteria as it currently stands. Therefore, we invite you to submit a revised version of the manuscript that addresses the points raised during the review process.

The ability to peer-review a paper is dependent on the clarity of writing. In this regard, two issues must be dealt with before resubmission and re-review. In this regard, the manuscript requires a major improvement in English writing style and in clarity with respect to specific ideas as indicated by Reviewer 1.

2, While the issue of novelty pointed out by Reviewer 2 is not relevant for this journal, the authors should nonetheless identify which of the findings add to our current knowledge in the field.

We look forward to receiving your revised manuscript.

Kind regards,

Arthur J. Lustig, PhD

Academic Editor

PLOS ONE

“This work was supported by the Jiangsu Provincial Special Program of Medical Science (BE2019750), National Natural Science Foundation of China (81827805), National Key Research and Development Program (2018YFA0704100, 2018YFA0704104), Science Fund for Creative Research Groups of the National Natural Science Foundation of China (61821002), and Interventional Radiology Scientific Research Special Fund Project of Jiangsu Medical Association (SYH-3201140-0025 (2021020)).”

Reviewers' comments:

Reviewer's Responses to Questions

**Comments to the Author**

1. Is the manuscript technically sound, and do the data support the conclusions?

Reviewer #1: Partly

Reviewer #2: Yes

2. Has the statistical analysis been performed appropriately and rigorously? 

Reviewer #1: Yes

Reviewer #2: Yes

3. Have the authors made all data underlying the findings in their manuscript fully available?

Reviewer #1: Yes

Reviewer #2: Yes

4. Is the manuscript presented in an intelligible fashion and written in standard English?

Reviewer #1: No

Reviewer #2: Yes

5. Review Comments to the Author

Reviewer #1: This study addressed the role of RRM2 in various tumors, which provide some interesting information for understanding carcinogenesis. There are some flaws needed to be addressed, as followings.

1- English should be carefully revised throughout the manuscript.

2- The quality of figures is very poor, and should be improved.

3- Introduction, page 3, line 22: “…to perform pan-cancer analysis of genes…” should be “…to perform a pan-cancer analysis of genes…”

4- Description of RRM2 gene expression should be in Italic and consistent through the manuscript.

5- Material and methods, page 4, line 78: “we chose…” should be “We chose…”

6- Material and methods, page 4, line 79: “…immune infiltrates of endothelial cell…” should be “…immune infiltrates of endothelial cells…”

7- Material and methods, page 5, line 97: “[24)” should be “[24]”

8- Material and methods, page 5, line 97: “pan-cencer” should be “pan-cancer”

9- Material and methods, page 4, “Genetic alteration analysis section”: In the analysis of RRM2 gene mutations, it is better to provide the accession number of RRM2 coding sequence used for the analysis.

10- Results, page 6, “RRM2 expression and clinical correlation in pan-cancer”: The authors examined the correlation between RRM2 expression and clinical characteristics of cancer patients, and found that RRM2 expression was significantly related to age, race, tumor stage, and status of cancer patients. It is better to describe in the main text, which age, race and tumor stage are associated with high RRM2 expression level.

11- Results, page 6, line 143-144: “…tumor patients’ clinical characteristics of patients with pan-cancer…” should be “…tumor clinical characteristics of patients with pan-cancer…”

12- Results, page 9, line 218: “…tumor mutational burden) TMB)…” should be “…tumor mutational burden (TMB)…”

13- Results, page 10, line 273: “…in Fig 7A We the…” should be “…in Fig 7A. We the…”

14- Discussion, page 13, lines 376-379: the authors found that “cell cycle”, “p53 signaling pathway”, “cell senescence, and other cellular pathways have potential roles in the effect of RRM2 on carcinogenesis. It is better to discuss these results according to the literature data.

Reviewer #2: Numerous experimental studies have been conducted to show that RRM2 is associated with the occurrence and development of tumors. And the function and mechanism of RRM2 in tumors has also been relatively well studied. Therefore, this manuscript lacks novelty.

6. PLOS authors have the option to publish the peer review history of their article (what does this mean?). If published, this will include your full peer review and any attached files.

Reviewer #1: No

Reviewer #2: No

---

## [Author Response · Author response to Decision Letter 0]

29 Jan 2024

Response to Reviewers

Reviewer #1: This study addressed the role of RRM2 in various tumors, which provide some interesting

information for understanding carcinogenesis. There are some flaws needed to be addressed, as followings.

1-English should be carefully revised throughout the manuscript.

A: I've done a native polish on the entire manuscript by Elsevier professional language polishing services.

2-The quality of figures is very poor, and should be improved.

A: All the figures have been improved.

3-Introduction, page 3, line 22: “…to perform pan-cancer analysis of genes…” should be “…to perform a pan-cancer analysis of genes…”

A: “…to perform pan-cancer analysis of genes…” has been modified into “…to perform pan-cancer analyses of genes…”according to Elsevier professional language polishing services.

4-Description of RRM2 gene expression should be in Italic and consistent through the manuscript.

A: All description of RRM2 gene expression have be revised in Italic and consistent through the manuscript.

5-Material and methods, page 4, line 78: “we chose…” should be “We chose…”

A: “we chose…” has been modified into “We chose…”

6-Material and methods, page 4, line 79: “…immune infiltrates of endothelial cell…” should be “…immune infiltrates of endothelial cells…”

A: …immune infiltrates of endothelial cell…” has been modified into “…immune infiltrates of endothelial cells…”

7-Material and methods, page 5, line 97: “[24)” should be “[24]”

A: “[24)” has been modified into “[24]”

8-Material and methods, page 5, line 97: “pan-cencer” should be “pan-cancer”

A: “pan-cencer” has been modified into “pan-cancer”

9-Material and methods, page 4, “Genetic alteration analysis section”: In the analysis of RRM2 gene mutations, it is better to provide the accession number of RRM2 coding sequence used for the analysis.

A: The number of RRM2 coding sequence is NC_000002.12NC_000002.12

10-Results, page 6, “RRM2 expression and clinical correlation in pan-cancer”: The authors examined the correlation between RRM2 expression and clinical characteristics of cancer patients, and found that RRM2 expression was significantly related to age, race, tumor stage, and status of cancer patients. It is better to describe in the main text, which age, race and tumor stage are associated with high RRM2 expression level.

A: The clinical characteristics of patients have been described in the main text, which age, race and tumor stage are associated with high RRM2 expression level.

11-Results, page 6, line 143-144: “…tumor patients’ clinical characteristics of patients with pan-cancer…” should be “…tumor clinical characteristics of patients with pan-cancer…”

A: “…tumor patients’ clinical characteristics of patients with pan-cancer…”have been modified into “…the clinical characteristics of patients with pan-cancer…”according to Elsevier professional language polishing services.

12-Results, page 9, line 218: “…tumor mutational burden) TMB) …” should be “…tumor mutational burden (TMB)…”

A: “…tumor mutational burden) TMB)…” has been modified into “…tumor mutational burden(TMB)…”

13-Results, page 10, line 273: “…in Fig 7A We the…” should be “…in Fig 7A. We the…”

A: “…in Fig 7A We the…” has been modified into “…in Fig 7A. We the…”

14-Discussion, page 13, lines 376-379: the authors found that “cell cycle”, “p53 signaling pathway”, “cell senescence, and other cellular pathways have potential roles in the effect of RRM2 on carcinogenesis. It is better to discuss these results according to the literature data.

A: The “cell cycle”, “p53 signaling pathway”, “cell senescence, and other cellular pathways have potential roles in the effect of RRM2 on carcinogenesis. I discussed these results according to the literature data.

Reviewer #2: Numerous experimental studies have been conducted to show that RRM2 is associated 

with the occurrence and development of tumors. And the function and mechanism of RRM2 in 

tumors have also been relatively well studied. Therefore, this manuscript lacks novelty.

A: Indeed, many studies have reported the close relationship between RRM2 and tumorigenesis and development, but most of them still focus on tumor invasion and proliferation and corresponding signaling pathways and other tumor molecular mechanisms. In this paper, we have made a relatively new interpretation of the mechanism of RRM2 regulation of tumor from the perspective of the immune mechanism of RRM2 and tumor-infiltrating lymphocytes and immune checkpoint molecules, which may provide certain theoretical support for tumor immunotherapy.

---

## [Decision Letter · Decision Letter 1]

14 Feb 2024

PONE-D-23-34369R1A Comprehensive Analysis of Ribonucleotide Reductase Subunit M2 for Carcinogenesis in Pan-CancerPLOS ONE

Dear Dr. Zeng,

Thank you for submitting your manuscript to PLOS ONE. After careful consideration, we feel that it has merit but does not fully meet PLOS ONE’s publication criteria as it currently stands. Therefore, we invite you to submit a revised version of the manuscript that addresses the issue raised during the review process.

 Specifically, respond to Reviewer 2 with regard to a clarification on the rationale for the enrichment analysis of SND1-related partners. However you can ignore the comments about the resolution issues since the downloaded figures are sufficiently resolved. Please submit your revised manuscript by Mar 30 2024 11:59PM. If you will need more time than this to complete your revisions, please reply to this message or contact the journal office at plosone@plos.org. Please include the following items when submitting your revised manuscript:A rebuttal letter that responds to each point raised by the academic editor and reviewer(s). You should upload this letter as a separate file labeled 'Response to Reviewers'.A marked-up copy of your manuscript that highlights changes made to the original version. You should upload this as a separate file labeled 'Revised Manuscript with Track Changes'.An unmarked version of your revised paper without tracked changes. You should upload this as a separate file labeled 'Manuscript'.If applicable, we recommend that you deposit your laboratory protocols in protocols.io to enhance the reproducibility of your results. Protocols.io assigns your protocol its own identifier (DOI) so that it can be cited independently in the future. For instructions see: https://journals.plos.org/plosone/s/submission-guidelines#loc-laboratory-protocols. Additionally, PLOS ONE offers an option for publishing peer-reviewed Lab Protocol articles, which describe protocols hosted on protocols.io. Read more information on sharing protocols at https://plos.org/protocols?utm_medium=editorial-email&utm_source=authorletters&utm_campaign=protocols.

We look forward to receiving your revised manuscript.

Kind regards,

Arthur J. Lustig, PhD

Academic Editor

PLOS ONE

Journal Requirements:

Reviewers' comments:

Reviewer's Responses to Questions

**Comments to the Author**

1. If the authors have adequately addressed your comments raised in a previous round of review and you feel that this manuscript is now acceptable for publication, you may indicate that here to bypass the “Comments to the Author” section, enter your conflict of interest statement in the “Confidential to Editor” section, and submit your "Accept" recommendation.

Reviewer #1: All comments have been addressed

Reviewer #2: (No Response)

2. Is the manuscript technically sound, and do the data support the conclusions?

Reviewer #1: Yes

Reviewer #2: Yes

3. Has the statistical analysis been performed appropriately and rigorously? 

Reviewer #1: Yes

Reviewer #2: Yes

4. Have the authors made all data underlying the findings in their manuscript fully available?

Reviewer #1: Yes

Reviewer #2: Yes

5. Is the manuscript presented in an intelligible fashion and written in standard English?

Reviewer #1: Yes

Reviewer #2: Yes

6. Review Comments to the Author

Reviewer #1: Authors addressed the comments raised by this reviewer and this paper seems to be now ready for publication.

Reviewer #2: 1. Why did "Enrichment analysis of SND1-related partners"? Is there a relationship of RRM2 and SND1?

2. Please consider to cite "" ext-link-type="uri" xlink:type="simple">https://doi.org/10.1016/j.gendis.2022.11.022".

3. All figures have to be uploaded in high resolution.

7. PLOS authors have the option to publish the peer review history of their article (what does this mean?). If published, this will include your full peer review and any attached files.

Reviewer #1: No

Reviewer #2: No

---

## [Author Response · Author response to Decision Letter 1]

17 Feb 2024

Reviewer #2: 

1. Why did "Enrichment analysis of SND1-related partners"? Is there a relationship of RRM2 and SND1?

A: I'm very sorry. Thank you for your careful review. This is actually a clerical error on my part. It's actually “Enrichment analysis of RRM2-related gene”. I have already corrected all the relevant clerical errors in the Methods and Results. 

2. Please consider to cite "https://doi.org/10.1016/j.gendis.2022.11.022".

A: Sorry, I don't know what this link means and where to reference it. Besides, this link is typed to show AN ERROR with DOI NOT FOUND.

3. All figures have to be uploaded in high resolution.

A: All figures have been uploaded in high resolution

---

## [Editor Report · Decision Letter 2]

21 Feb 2024

A Comprehensive Analysis of Ribonucleotide Reductase Subunit M2 for Carcinogenesis in Pan-Cancer

PONE-D-23-34369R2

Dear Dr. Zeng,

We’re pleased to inform you that your manuscript has been judged scientifically suitable for publication and will be formally accepted for publication once it meets all outstanding technical requirements.

Kind regards,

Arthur J. Lustig, PhD

Academic Editor

PLOS ONE
---

## [Editor Report · Acceptance letter]

21 Mar 2024

PONE-D-23-34369R2 

PLOS ONE

Dear Dr. Zeng, 

I'm pleased to inform you that your manuscript has been deemed suitable for publication in PLOS ONE. Congratulations! Your manuscript is now being handed over to our production team.

Kind regards, 

on behalf of

Dr. Arthur J. Lustig 

Academic Editor

PLOS ONE